# In Vitro Evaluation of Winged Bean (*Psophocarpus tetragonolobus*) Tubers as an Alternative Feed for Ruminants

**DOI:** 10.3390/ani13040677

**Published:** 2023-02-15

**Authors:** Chanon Suntara, Napudsawun Sombuddee, Saowalak Lukbun, Natdanai Kanakai, Pachara Srichompoo, Sompong Chankaew, Benjamad Khonkhaeng, Pongsatorn Gunun, Nirawan Gunun, Sineenart Polyorach, Suban Foiklang, Anusorn Cherdthong

**Affiliations:** 1Department of Animal Science, Faculty of Agriculture, Khon Kaen University, Khon Kaen 40002, Thailand; 2Department of Agronomy, Faculty of Agriculture, Khon Kaen University, Khon Kaen 40002, Thailand; 3Department of Agricultural Innovation and Technology, Institute of Interdisciplinary Studies, Rajamangala University of Technology Isan, Nakhon Ratchasima Campus, Nakhon Ratchasima 30000, Thailand; 4Department of Animal Science, Faculty of Natural Resources, Rajamangala University of Technology-Isan, Sakon Nakhon Campus, Phangkhon, Sakon Nakhon 47160, Thailand; 5Program in Animal Production Technology, Faculty of Technology, Udon Thani Rajabhat University, Udon Thani 41000, Thailand; 6Department of Animal Production Technology and Fisheries, Faculty of Agricultural Technology, King Mongkut’s Institute of Technology Ladkrabang, Bangkok 10520, Thailand; 7Faculty of Animal Science and Technology, Maejo University, Chiangmai 50290, Thailand

**Keywords:** winged bean tubers, roughage, in vitro gas production, degradability, alternative energy

## Abstract

**Simple Summary:**

Cassava is becoming increasingly popular in the worldwide market, thus leading to price surges. This phenomenon has a direct influence on ruminant farming, as cassava is used as an energy source in more than half of their meals. Accordingly, we have been searching for an alternative feed to compensate for this uncertainty and, more specifically, for tuberous crops such as winged bean tubers (WBTs) that have the potential to deliver nutrition equivalent to or even greater than conventional cassava chips. It was hypothesized that WBTs would be ideal for usage as animal feed and a unique alternative feedstock for ruminants. We identified that, when associated with grass, WBTs may be effectively utilized to substitute cassava chips without impairing rumen function. To summarize, even though it is too early to determine if WBTs can be utilized as a substitute for cassava, there are aspects of our study that indicate this possibility, and further research is required to evaluate the practicability of this alternative feedstuff.

**Abstract:**

The purpose of the current study is to determine the effects of the replacement of cassava chips with winged bean (*Psophocarpus tetragonolobus*) tubers (WBTs) on gas production parameters, in vitro degradability, and ruminal fermentation in ruminant diets. The study was performed using a 3 × 4 factorial arrangements and was designed using a completely random method. Factor A employed three various roughage sources that were frequently used by locals to feed ruminants: *Oryza sativa* L. (a1), *Brachiaria ruziziensis* (a2), and *Pennisetum purpureum* (a3). The levels of WBTs substitutions for cassava chips at 0%, 33%, 66%, and 100% in the diet were selected as factor B. The experiment’s findings revealed that replacing the cassava chips in the diet with WBTs at levels of 66 and 100% enhanced the fermentation process by producing a high gas volume at 96 h when Ruzi grass (RZ) was used as the main source of roughage (*p* < 0.01). The interaction between the roughage source and WBTs showed that organic matter (OM) degradability improved markedly in the case of RZ grass in combination with WBTs at all levels. Both the total volatile fatty acids (TVFAs) at 8 h of incubation and the average value decreased when a complete substitution of casava chips with WBT (WBT 100%) was employed or when employing rice straw as the main source of roughage (*p* < 0.01). There was no interaction between WBTs and roughage source on the ammonia–nitrogen (NH_3_-N) concentration (ml/dL) and rumen microbial count (*p* > 0.05). In summary, WBTs can be used effectively when combined with grass (Ruzi and Napier). The implementation of WBTs as a novel alternative feed may effectively replace cassava chips without affecting rumen function.

## 1. Introduction

The unpredictability of current agricultural production should be considered in a variety of global contexts [1]. The problems faced by feed production have been demonstrated by countless conflicts [2]. The conflict that erupted between the world’s two largest exporters of feed (Russia and Ukraine) has further affected the security of the global animal feed supply [3]. Thailand, which imports a sizable amount of feed ingredients, is severely impacted by this crisis. The Thai Feed Mill Association (TFMs) [4] indicated that prices for animal feed in Thailand, such as those of maize, wheat, and barley, increased by 30.5%, 56.6%, and 52.9%, respectively, between 2021 and 2022. In a number of developing countries, the insecurity of animal feed is aggravated by the inaccessibility of inexpensive animal feed. However, feed price specialists have noted that despite the fact that the Russia–Ukraine conflict has led to a rise in short-term pricing, it presents an opportunity for a long-term shift toward feed resources [5].

Cassava chips (*Manihot esculenta*) are tuberous plants that are extensively grown as the primary feed source for ruminant animals in Thailand [6]. However, this plant is also used in the human food industry, which has led to opportunities for competition between cassava intended for human use and as feed that could, in turn, lead to unstable prices in the future. In addition, 64% of Thailand’s cassava is exported in the form of a variety of products (cassava chips, cassava pellets, modified cassava starch, and other types of cassava), with cassava chips being one of the most well-known [7]. According to Singvejsakul et al. [8], Thailand is the top exporter of cassava products globally, with European and Asian markets serving as the two most significant export markets. As a result of the Russia–Ukraine crisis, the increasing demand for cassava in the international market and its displacement of rice, potatoes, wheat, and corn as one of the most important products for the world economy have caused its price to rise by 20% [4]. In addition, with the challenges that Thailand has been facing because of the COVID-19 pandemic, the problem of climate change, international transportation issues, etc., all these factors increase the price of cassava chips. Therefore, introducing new species of tuberous plants or other plants can help increase the variety of local feed and minimize feed shortages in this country.

*Psophocarpus tetragonolobus* (L.) DC. is underused but has a wide genetic potential with respect to its use as a source of high-quality feed and fodder [9]. This plant can be grown well in humid tropical environments, with tuber yields ranging from 15.2 to 15.5 T/ha (in comparison to a cassava chip production of approximately 20 ton/h) [10,11]. This plant is a high-nutritional-value tropical legume vegetable [6] and, despite having a smaller yield, this plant is thought to be more nutritious than cassava chips. Its young pods, seeds, leaves, flowers, shoots, and tubers can be used as animal feed [12]. Its tubers have high levels of crude protein (CP) and carbohydrates (on a dry basis) of 3–15% and 34–40%, respectively [9,13]. Currently, winged bean tubers are genetically enhanced to grow exponentially and yield more tubers [14]. However, there is currently a lack of information about the winged bean tuber, and there is scarce information available regarding its nutritional advantages [10]. With respect to investigating alternative feed for use in Thailand, ruminant utilization data represent both an attractive topic and a research gap. Consequently, the purpose of the present study is to evaluate the impacts of replacing cassava chips with winged bean tubers on gas production parameters, in vitro degradability, and ruminal fermentation in ruminant diets. We established the hypothesis that winged bean tubers would be ideal for use as both animal feed and a novel feedstock for ruminants in the area due to the variable price of feed.

## 2. Materials and Methods

The animal care and use committee of Khon Kaen University provided their approval for all the methods and procedures used in this study (record no. IACUC-KKU 38/62).

### 2.1. Plant Material and Growth Conditions

Winged bean (*Psophocarpus tetragonolobus* (L.) DC.) tubers (WBT) were obtained from Dr. Chankaew (Department of Agronomy, KKU). The WBT were grown in plots of 5 × 1 m, with 1 m between rows, 0.5 m between plants, and with each plot containing ten plants, and there was a 2 m gap between each plot within the same row. A total of two applications of fertilizer were performed: one at 21 days after planting (14.06 kg/ha, N_2_-P_2_O_5_-K_2_O) and another at two months of age (18.75 kgN_2_/ha, 37.50 kgP2O5/ha, and 18.75 kgK_2_O/ha) [10]. Manual weed control was routinely employed as the crop grew. In addition, plants received regular irrigation, and disease and pest issues were dealt with as necessary. Tubers were removed from the plots at the age of eight months, and their fresh weights were immediately recorded. Fresh tubers of the winged bean accessions from each plot were immediately sub-sampled, rinsed with tap water, sliced into small chips, and dried in a forced-air drying oven at 55 °C for 72 h, or until consistent moisture was achieved. A grinder was used to grind the samples into a powder (Wiley Mill, Arthur H. Thomas Co., Philadelphia, PA, USA). To obtain a representative sample for chemical analysis, the samples were mashed into a powder and sieved through a 1.0 mm mesh screen.

### 2.2. Treatments and Experimental Design

The study was performed using a 3 × 4 factorial scheme and was designed using a completely random method. Factor A employed three various roughage sources that were frequently used by locals to feed ruminants: *Oryza sativa* L. (a1), *Brachiaria ruziziensis* (a2), and *Pennisetum purpureum* (a3). The levels of WBT substitutions for cassava chips at 0%, 33%, 66%, and 100% in the diet were selected as factor B, and these levels represent b1, b2, b3, and b4, respectively. The NRC [15] was used to designate animal feed according to component composition. The ingredient composition and chemical composition of the basal diet are shown in Table 1 and Table 2.

### 2.3. Rumen Liquid Sampling

In the morning, before feeding, rumen fluid was collected from two dairy steers via a rumen fistula. The average body weight of these steers was 350 ± 10.0 kg. The steers were fed rice straw as a roughage source and supplemented with a concentrated diet (16% CP and 69% TDN) that was formulated to meet their requirements before collecting rumen fluid [15]. In order to perform rumen cannula sampling, the cannular lid was removed, and four different parts of the rumen were then sampled: the ventral sac, the atrium or reticulum, two samples from the feed mat, and the atrium. Rumen fluid from the two steers was collected and combined. About 1.5 L of rumen fluid was strained through four sheets of cheesecloth, placed in a thermally insulated container (kept at 39 °C), and then transported to the laboratory within a period of 15 min.

### 2.4. In Vitro Rumen Incubation and Gas Determination

The in vitro analysis and the preparation of the artificial saliva and rumen fluid followed the procedures described by Menke [16]. In an anaerobic environment, rumen medium preparations including distilled water (1095 mL), a trace mineral mixture (0.23 mL), a macro mineral mixture (365 mL), a resazurine mixture (1 mL), a reduction mixture (60 mL), and a buffer mixture (730 mL) were combined with rumen liquor (660 mL). The ground WBT samples were weighed (at 500 mg) in 50 mL bottles at their respective levels of total substrate. A 50 mL artificial inoculum was removed and injected into the serum bottles containing the substrate for their separate treatments. Each treatment bottle contained 40 mL of the artificial saliva plus rumen fluid. Synthetic rubber stoppers were used on all experimental bottles with crimped aluminum seals and were kept in storage at 39 °C. Three bottles were used for each treatment to measure gas kinetics and gas production. Gas production was assessed at 0, 0.5, 1, 2, 3, 4, 5, 6, 7, 8, 9, 10, 11, 12, 18, 24, 48, 72, and 96 h of incubation by gently swirling the substance every 3 h throughout the incubation time and measuring gas production with a 20 cc glass precision syringe. Only rumen inoculum was present in the blank bottles, and the net gas production was calculated by subtracting the average value of gas production from the experimental bottles containing blanks. At 4 and 8 h, the serum bottles were opened, and the contents were analyzed in terms of pH, while volatile fatty acids (VFA) were analyzed using GC, NH_3_-N (Kjeldahl methods; [17]), and microbial counts (Boeckel & Co. GmbH & Co., Hamburg, Germany). Nutrient degradability was measured, and the incubation residues were collected in an Ankom filter bag (ANKOM 200, ANKOM Technology, New York, NY, USA).

### 2.5. Chemical Analysis and Calculations

All analyses were performed in the animal nutritional laboratory at the Department of Animal Science, Faculty of Agriculture, Khon Kaen University (KKU), Khon Kaen (16°26′48.16″ N, 102°49′58.8″ E), Thailand. The DM content of the feed was determined by first drying the feed ingredient (roughage and WBT) at 60 °C overnight, grinding it, and then drying it once more at 60 °C overnight, according to the method reported by Karlsson et al. [18]. The DM content of the concentrate was determined by drying it at 105 °C overnight. The feed samples were chemically analyzed along with the AOAC method, including in terms of the percentage of ash, which was determined by ignition at 550 °C for 3 h (Ash; ID 492.05), and ether extract, which was determined and set up by the apparatus of Soxhlet extraction, including and using petroleum ether as solvent extraction (EE; ID 445.08). An examination of CP was performed to determine the quantity of this component (Leco FP828 Nitrogen Analyzer, LECO Corporation, Saint Joseph, MI, USA). Neutral detergent fiber (NDF), acid detergent fiber (ADF), and acid detergent lignin (ADL) fractions were determined using a detergent analysis method developed by Van Soest et al. [19]. Hemicellulose was calculated as NDF − ADF and cellulose as ADF − ADL [20]. The IVDMD, IVOMD, and IVNDFD (% of NDF) were determined by the procedures of Goering [21]; IVDMD, IVOMD, and IVNDFD were calculated as follows:

IVDMD = (DM incubated (g) − DM residue (g)) × 100%/DM incubated (g)

IVOMD = (OM incubated (g) − OM residue (g)) × 100%/OM incubated (g)

IVNDFD = (NDF incubated (g) − NDF residue (g)) × 100%/NDF incubated (g)

According to Fawcett [22], a spectrophotometer was used to quantify ammonia nitrogen (NH_3_-N) (UV/VIS Spectrophotometer, PG Instruments Ltd., London, UK). Individual VFA concentrations were analyzed by GC (Model HP 6890, Hewlett-Packard Co., Ltd., New York, NY, USA) according to the method described by Kozaki [23].

### 2.6. Statistical Analysis

The kinetics of ruminal fermentation proposed by Ørskov and McDonald [24] were estimated using mathematical non-linear models for recording gas production, and the data concerning the produced gas were entered into the equation:Y = a + b (1 − exp^(−ct)^)
where a = volume of gas produced from soluble fraction, b = volume of gas produced from insoluble fraction, c = gas production rate constant for insoluble fraction, t = incubation time, a + b = potential extent of gas production, and Y = gas produced at time t

Data regarding chemical composition, microorganism counts, in vitro degradability, and gas kinetics of the feed sample were examined. This study was performed using a 3 × 4 factorial arrangements (Roughage source × WBT inclusion), to which ANOVA was applied using general linear model (GLM) procedures via SAS software [25], and was designed using a completely random method. The average treatment value was calculated using the SAS program’s least-square means (LSMEANS) option (SAS Institute Inc., Version 6.2.9200, Cary, NC, USA), with statistical modeling as follows:Yij = µ + αi + βj + αβij + εij
where Yij = observation, µ = overall mean, αi = roughage source effect (I = *Oryza sativa* L., *Brachiaria ruziziensis*, and *Pennisetum purpureum*), βj = level of fermented WBT replacing cassava chip (j = WBT at 0%, 33%, 66% and 100% levels), αβij = roughage source × levels of WBT, and εij = error. Duncan’s new multiple range test (DMRT) was used to determine the differences among the means, which were accepted at *p* < 0.05 [26].

## 3. Results

### 3.1. Ingredient Composition and Chemical Composition

The ingredient composition (as content in the total ingredient) and chemical composition of the basal diet of winged bean tubers, rice straw, Ruzi grass, and Napier grass are shown in Table 1. Winged bean tuber (WBT) has a DM content of 435 g kg^−1^, as well as 4.5 g kg^−1^ EE, 189 g kg^−1^ CP, 168 g kg^−1^ NDF, 11.5 g kg^−1^ Ca, and 7.9 g kg^−1^ P on a DM basis. The roughage sources, Ruzi and Napier grass, had high CP (130 and 112 g kg^−1^, respectively), EE (19.2 and 20.5 g kg^−1^ DM, respectively), and low P content (1.67 and 1.80 g kg^−1^ respectively). Conversely, rice straw had low CP (33.7 g kg^−1^ DM) and EE (7.5 g kg^−1^ DM) content along with a high P content (2.31 g kg^−1^ DM) compared to other roughage. The NDF of the roughage sources varied, ranging from 538 g kg^−1^ DM in Napier grass to 672 g kg^−1^ DM in rice straw. The main source of energy in terms of the ingredients’ composition is cassava chips (which range from 0–500 g kg^−1^), which have been replaced by 33%, 66%, and 100% WBT (Table 2). The concentrate diet contained 953 to 967 g kg^−1^ DM, and the nutrients varied from 952 to 921 g kg^−1^ OM and 42.3 to 38.9 g kg^−1^ EE, while the rice straw group showed slightly lower OM and EE levels than the grass group. Different roughage types and WBT levels were employed to supply diets and to balance protein levels with urea to ensure an equivalent supply of nitrogen in the experimental diets (158 to 162 g kg^−1^ CP). The NDF and ADF levels of rice straw were high: 389 to 395 g kg^−1^ and 145 to 155 g kg^−1^, respectively.

### 3.2. Cumulation and Kinetics of Gas

As the incubation time increased, cumulative gas production (Figure 1) continuously increased until becoming relatively stable after 60 h. An interaction between roughage sources and WBT levels was observed with respect to the level of cumulative gas production at 96 h (*p* < 0.01). Replacing the cassava chips in the diet with WBT at levels 66 and 100% enhanced the fermentation process by producing a high gas volume at 96 h when Ruzi grass was used as the main source of roughage (*p* < 0.01). Regarding the kinetics of gas production (Table 3), the gas production from the immediately soluble fraction (Gas a) and the gas production rate constant for the insoluble fraction (Gas c) were not affected by the interaction of the roughage source or WBT levels, but the interaction impact increased the gas production level from the insoluble fraction (Gas b; *p* < 0.01) and the gas potential extent of gas production (Gas a + b; *p* < 0.05). The amount of Gas a is lower in the RZ group than it is with NP and RS groups, while the RS group reduced the amount of Gas c by 13% (*p* < 0.01). A comparison between grass and rice straw showed that grass provided greater amounts of Gas b, Gas c, and Gas a + b, whereas the amount of Gas a was lower (*p* < 0.01).

### 3.3. Dry Matter, Organic Matter, and Fiber Degradability

There were no interactions between roughage source and WBT levels (*p* > 0.05; Table 4); however, the degradability of rice straw in DM decreased by 8.0% and 11.7% at 12 and 24 h of incubation, respectively (*p* < 0.05; Figure 2). There is no interaction between the roughage source and WBT levels showed that OM degradability improved markedly in the case of RZ grass in combination with WBT at all levels (625.1 to 682.9 g kg^−1^). The degradability of NDF was remarkable in terms of orthogonal contrast; when grass was applied, there was a 4.2% increase in NDF degradability over RS. However, the increase in NDF degradability only appeared at the 12 h incubation timepoint (*p* < 0.01), and there was no difference at the 24 h incubation timepoint (*p* < 0.05).

### 3.4. Ammonia–N and Volatile Fatty Acid Concentrations

The effects of the replacement of cassava chips with winged bean tubers on different roughage sources with respect to the ammonia–nitrogen concentration (NH_3_-N) and total volatile fatty acids (TVFAs) are shown in Table 5, and the proportion of rumen VFAs displayed significant differences, as reported in Table 6. No statistical difference in NH_3_-N production (ml/dL) was observed among the treatments (*p* > 0.05). Both the TVFA at 8 h of incubation and the average value decreased when completely replacing casava chips with WBT (WBT 100%) or when employing rice straw as the main source of roughage (*p* < 0.01). The acetate and butyrate concentrations (mol/100 mol) and the ratios of acetate to propionate are higher in the diet utilizing RS as the main source of roughage (*p* < 0.01), whereas the propionate content is lower (*p* < 0.01). The propionate content significantly decreased linearly after 8 h of incubation when the amount of WBT employed as a substitute for the cassava chip was increased (*p* < 0.01). An interaction between the roughage source and WBT was observed for butyrate at 8 h *(p* < 0.01). The use of RS as the main roughage source in combination with a WBT proportion of 66 and 100% in place of cassava chips produced the highest butyrate content alongside the use of RZ in combination with 100% WBT (*p* < 0.01). However, the RZ group was able to produce the least butyrate when combined with a 0% WBT proportion (*p* < 0.01).

### 3.5. Rumen Microbial Count

The effects of the replacement of cassava chips with winged bean tubers on different roughage sources with respect to the bacterial, protozoal, and fungal zoospore populations are shown in Table 7. There is no interaction between the roughage source and WBT levels concerning the number of microbial populations in the rumen (*p* > 0.05). The cassava chips were substituted with WBT at 66% and 100%, which linearly reduced the number of bacteria at 8 h of incubation by 3.3% and 4.4%, respectively (linear; *p* < 0.01). At 8 h of incubation time, the fungal zoospore population increased when the substitution level of WBT was raised to 66% before falling again when used at 100% (Quadratic *p* = 0.02).

## 4. Discussions

The nutritive value of feedstuffs is commonly determined by analyzing the level of in vitro gas production [27]. Such findings are accurately representative of the rumen since the in vitro gas production system is similar to the conditions in the rumen [28]. The exponential curve of in vitro cumulative gas production that was affected by the use of different roughage sources with WBT showed a significant increase in line with the above pattern in the RZ grass with WBT added (at 66% and 100%). An increase in the fermentable portion in feed can result in increased cumulative gas production [29]. As a result, an increase in gas production with specific treatments reflects an increase in substrate fermentability in the inoculum batch. This positive effect only occurs in the RZ group and not in the NP group, which is chemically similar to the former. As a result, it is important to note any differences that we have identified aside from those regarding chemical composition. Measures that may impact total gas production are still being researched. The highest values of gas production parameters in the RZ grass with WBT added (at 66% and 100%) are at least partly due to the difference in fermentable carbohydrates present in the structure of the roughage feed with WBT and the higher degradability of the insoluble fraction. In this study, when RZ grass was mixed with both levels of WBT (66% and 100%), a positive interaction effect on the gas kinetics pattern was observed, indicating that WBT can be expected to benefit rumen digestion. There are several factors that may contribute to the improved level of Gas b, and this change often affects Gas a + b as well. The insoluble fraction refers to the portion of feed that is not broken down and absorbed in the rumen but is fermentable in the lower parts of the digestive tract for further digestion and absorption [30]. When the insoluble fraction reaches the lower parts of the digestive tract, it is exposed to various types of microbes that attach to a variety of nutrients, including insoluble feed materials [31], and can produce gas as they break them down. Chumpawadee et al. [32] anticipated that lignin in animal feed would increase Gas b value. This should have made the RS treatment group in this experiment produce more Gas b than the other groups, but our results did not reveal this. Lignin is a non-carbohydrate fraction that is difficult for microbes to decompose; therefore, gas produced from this fraction may have no effect on the level of Gas b. This reasoning was demonstrated by the experiments of Sommart et al. [33] that compared the kinetics of gas production; it was found that RZ produces Gas b at a level of 136.7 mL/0.5 g substrate (in our experiment, RZ produced Gas b at a level of 139.9 mL/0.5 g substrate), while RS produces Gas b at only 122.7 mL/0.5 g substrate, in which RS’s lignin content is much higher than that of RZ. If lignin actually affects Gas b, it should have been demonstrated that RS has a greater Gas b value. However, this experiment suggests that adding an organic fermentable substance such as urea to RS improves the insoluble fraction of RS (144.2 mL/0.5 g substrate). Olivera [34] showed that with a small amount of an easily soluble fraction in the feed, the utilization of the insoluble fraction was poor, resulting in a low nutritional value and a low level of Gas b produced.

The increase in the activity of fiber-utilizing bacteria in the rumen may lead to the promotion of Gas b or Gas a + b. Higher fermentation of the insoluble fraction was observed in the experiments conducted by Suntara et al. [35], where molasses, urea, and yeast were used. Yeast can stimulate fiber-utilizing bacteria in the rumen and appears to have executed this admirably, resulting in an 11.5% increase in Gas b when compared to the RS control group. The cell wall structure of the roughage being exploited may also cause a change in Gas b. This experiment suggests that hemicellulose may also assist in boosting the amount of Gas b. Akinfemi et al. [36] compared different tropical feed sources and by-products; it was shown that fermented sorghum waste produces greater levels of Gas b than other feed sources. This study’s results were remarkable when fermented sorghum waste was enriched in hemicellulose. Previous research has shown that the hemicellulose content influences not only Gas b but also Gas a + b. Olfaz et al. [36] presented significant data demonstrating that *Morus alba* L. (Mulberry, 29.1% hemicellulose) contains hemicellulose in a higher proportion than other plants studied in comparison. This plant can produce 30.2% more Gas a + b than *Olea europaea* L. (Olive) and *Citrus aurantium* L. (Bitter Orange). According to the data we have obtained, Ruzi grass may contain a higher concentration of certain compounds, especially hemicelluloses (287 g/kg DM^−1^), and WBT may have easily fermentable substances that promote the digestion of feed ingredients; this combination is more easily fermented by rumen microbes and produces more gas as a byproduct. Furthermore, our findings indicate that roughage source improvement, e.g., in terms of RS, is still required, and the interaction of WBT with RS did not increase the degradability of the insoluble fraction of Gas b or Gas a + b.

In addition, the roughage source has a significant impact on the alteration of gas production from the immediately soluble fraction (Gas a). The mean values of Gas a were negative during the incubation of the different roughage sources with winged bean tubers replacing cassava chips, particularly with respect to RZ, which showed more negative values than the other groups (Table 3). Negative Gas a values have also been observed in the in vitro gas research of Sommart et al. [33], wherein RZ yielded Gas a at −28.3 mL/0.5 g substrate (our experiment RZ: Gas a −20.22 mL/0.5 g substrate), while RS yielded Gas a at −17.5 mL/0.5 g substrate (our experiment RS: Gas a −14.69 mL/0.5 g substrate), suggesting that roughage feed containing more soluble parts may produce more negative values. These findings suggested that a lag phase may occur in the early stages of incubation due to delays in the microbial colonization of the substrate. Microbial colonization can cause a delay in the onset of fermentation; alternatively, a lag can occur after the soluble portion of the substrate has been consumed but the fermentation of the cell walls has not yet begun [37]. The use of forage as a roughage source had a significant influence on the rate constant for the insoluble fraction (Gas c). The presence of an easily digestible cell structure in the feed or the improvement of raw materials through the use of easily fermentable additives had the potential to increase the rate of gas production [38]. Thus, forage, with its easily digestible cell wall structure, may be more easily utilized than RS, resulting in the highest rate of fermentation (forage 0.78 mL/h vs. rice straw 0.68 mL/h). This is in accordance with the research conducted by Sommart et al. [33], who found that using RZ grass as a roughage source resulted in higher Gas c levels than using RS (0.052 mL/h vs. 0.035 mL/h, respectively). In accordance with the findings of Ekani and Wahyono [39], no improvement in rice straw produced lower Gas c levels than the improvement group. RS is classified as roughage feed of low quality. It has a high fiber content and degrades slowly. When compared to other forages, these characteristics show poor utilization efficiency in in vitro gas and degradability experiments.

The in vitro dietary dry matter degradability responded positively to forage, whereas RS was poor. Our observations are similar to those made in Yulistiani et al.’s [40] trial, which compared RS to NP. The IVDMD of RS was 59.1% lower than that of NP, which was 63.9%. There are several factors that contribute to this difference in degradability. One factor is the chemical composition of the two types of feed. Grass is generally higher in nutrients such as protein and minerals, which stimulate rumen microbes [41]. In contrast, RS is lower in these nutrients and is more fibrous, which makes it more difficult to digest [42]. Another factor is the structure of the feed. Grass contains less fiber and lignin than rice straw, which may explain why it is easier for rumen microbes to break down and utilize. Rice straw, on the other hand, is more rigid, and has a more complex structure, which can make it more difficult to break down [43]. Finally, the presence of certain types of plant compounds, such as lignin in our study (the lignin content in grass was found to be three times less than in RS), can also affect the degradability of grass and rice straw. Lignin is found in higher concentrations in rice straw. The high lignin content of straw also contributes to its low digestion (<50%) [44]. Overall, the combination of these factors contributes to the generally higher degradability of grass in the rumen compared to rice straw. WBT in combination with grass (as roughage) had outstanding in vitro OM degradability, and the influence of forage could increase digestion efficiency better than the RS group. However, only WBT had no effect on the changes in feed degradability. Our findings are consistent with those of Suntara et al. [12], who used WBT in a concentrated diet. The replacement of cassava chips with WBT at 50% and 100% did not affect feed digestion changes in the experiments. The higher level of gas production in the grass-based diet was accompanied by a higher IVOMD. When compared to the data obtained using the nylon bag technique, previous research has shown that the kinetics of gas production are closely related to degradability and feed intake [45]. The strong relationship between the rate of DM or OM degradability and gas production has been documented previously [33,46]. The present experiment’s results are consistent with these observations.

VFAs play a key role in the digestion and metabolism of ruminant animals and are important for the overall health and productivity of these animals [47]. The total VFAs in the rumen can vary depending on a variety of factors, including the type of feed being consumed. In particular, when comparing RS and grass in this trial, it is likely that the VFA concentration in grass was improved. Rice straw is not a highly digestible feed or a good source of energy for animals [48]. As a result, when rice straw is consumed, the VFA concentration in the rumen is likely to be lower than when grass is consumed. The effect of WBT appears to have resulted in a decrease in the total VFAs in the formula. According to Suntara et al. [12], replacing WBT with cassava chips may lower the level of TVFAs by up to 20%. The causes of this occurrence are unknown. However, the chemical composition of WBT, which have a smaller percentage of starch than cassava, may result in reduced TVFA production potential. Carbohydrates are metabolized in the rumen by a variety of bacteria and enzymes before being converted into VFAs. All of these activities are carried out by rumen microbial enzymes in a succession of actions [49]. In our study, using grass as a roughage source resulted in an increase in the level of propionic acid (C3) compared to RS; however, high levels of WBT replacement decreased the proportion of C3. There are several factors that can cause an increase in the proportion of C3 in the rumen. The type and quality of the feed that an animal consumes can affect the production of C3 in the rumen [50]. Diets high in starch may lead to an increase in propionic acid production [51,52].

## 5. Conclusions

In conclusion, WBT can be utilized effectively when combined with grass (Ruzi and Napier). Ruzi grass demonstrated high gas production and degradability in vitro. However, the WBT proved ineffective in terms of gas production when combined with rice straw. Overall, the introduction of WBT as a novel alternative feed may effectively replace cassava chips without impairing rumen function. Further assessments of in vivo trials, including roughage sources and WBT interaction, are necessary.

## Figures and Tables

**Figure 1 animals-13-00677-f001:**
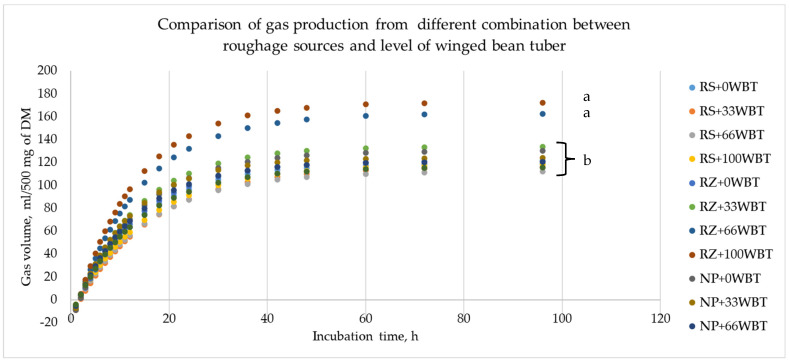
Cumulative gas production curves using observed data fitted by the gas production model for the combination of roughage sources and level of replacement of cassava chips with winged bean tubers in the concentrate diet throughout the incubation period (0 to 96 h). RS: rice straw; RZ: Ruzi grass; NP: Napier grass; 0WBT: 0% replacement of cassava with winged bean tubers; 33WBT: 33% replacement of cassava chips with winged bean tubers; 66WBT: 66% replacement of cassava chips with winged bean tubers; 100WBT: 100% replacement of cassava chips with winged bean tubers (roughage source × level of WBT *p* < 0.01, SEM = 7.79). Significant differences are indicated by different letters a and b.

**Figure 2 animals-13-00677-f002:**
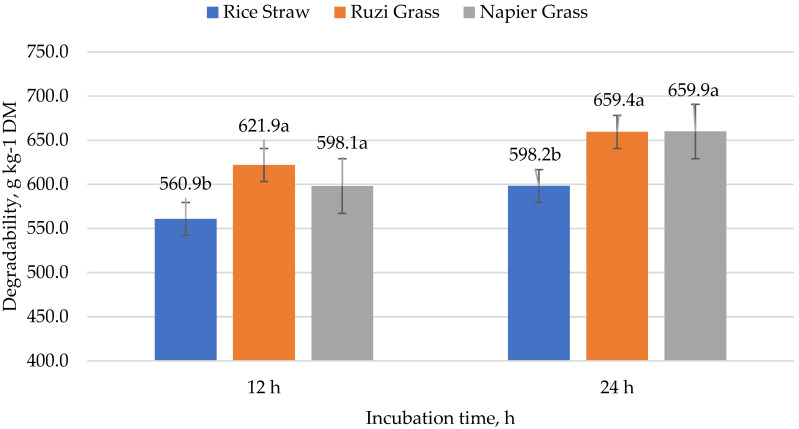
The in vitro dry matter degradability of diets in response to different roughage sources. All values are mean ± SEM. ^a,b^, Means that are sharing different superscripts are different (*p* < 0.05) within the same incubation stage.

**Table 1 animals-13-00677-t001:** Ingredient composition (as content in total ingredient) and chemical composition of the basal diet, winged bean tuber, rice straw, Ruzi grass, and Napier grass.

Chemical Compositions	Winged Bean Tuber	Rice Straw	Ruzi Grass	Napier Grass
Dry matter, g kg^−1^ fresh matter	435	861	234	192
g kg^−1^ dry matter				
Organic matter	963	859	893	886
Crude protein	189.8	33.7	131.8	112.3
Ether extract	4.5	7.5	19.2	20.5
Neutral detergent fiber (NDF)	168.5	672.1	592.6	538.3
Acid detergent fiber (ADF)	54.2	458.7	305.1	301.1
Acid detergent lignin (ADL)	5.1	90.5	31.6	35.2
Hemicellulose ^1^	113.8	214.2	287.3	237.9
Cellulose ^2^	49.1	368	273	265.8
Calcium	11.5	4.2	4.3	5.3
Phosphorus	7.9	2.3	1.7	1.8

^1^ Hemicellulose was calculated as NDF − ADF; ^2^ Cellulose was calculated as ADF − ADL.

**Table 2 animals-13-00677-t002:** Ingredients and chemical composition analysis (g/kg, dry matter (DM) basis) of substrates regarding in vitro gas production for evaluated roughage sources and replacement of cassava chips with winged bean tubers.

Item	Rice Straw				Ruzi Grass				Napier Grass			
	WBT 0	WBT 33	WBT 66	WBT 100	WBT 0	WBT 33	WBT 66	WBT 100	WBT 0	WBT 33	WBT 66	WBT 100
Ingredients, g kg^−1^												
Roughage												
Rice straw	200	200	200	200	-	-	-	-	-	-	-	-
Ruzi grass	-	-	-	-	200	200	200	200	-	-	-	-
Napier grass	-	-	-	-	-	-	-	-	200	200	200	200
Concentrate												
Cassava chip	500	335	170	0	500	335	170	0	500	335	170	0
Winged bean tuber	0	165	330	500	0	165	330	500	0	165	330	500
Soybean meal	95	95	95	95	95	95	95	95	95	95	95	95
Palm kernel meal	50	50	50	50	50	50	50	50	50	50	50	50
Rice bran	78	87	97	107	82	89	99	107	81	87	98	107
Palm oil	30	30	30	30	30	30	30	30	30	30	30	30
Salt	10	10	10	10	10	10	10	10	10	10	10	10
Dicalcium phosphate	5	5	5	5	5	5	5	5	5	5	5	5
Urea	29	20	10	0	25	18	8	0	26	20	9	0
Mineral and vitamin mix ^1^	3	3	3	3	3	3	3	3	3	3	3	3
Chemical composition												
Dry matter, g/kg^−1^	967	966	953	958	962	967	965	961	962	959	964	967
g kg^−1^ dry matter												
Organic matter	932	930	925	921	945	947	943	952	941	945	942	949
Crude protein	158	159	159	160	160	161	162	160	160	159	160	161
Ether extract	38.9	39.1	39.4	39.5	40.2	42.1	41.5	42.3	41.4	41.3	41.9	41.2
Neutral detergent fiber	389	392	393	395	373	375	376	379	369	372	377	378
Acid detergent fiber	145	147	150	155	138	142	139	140	137	140	136	139

^1^ Minerals and vitamins (each kg contains): Vitamin A: 10,000,000 IU; Vitamin E: 70,000 IU; Vitamin D: 1,600,000 IU; Fe: 50 g; Zn: 40 g; Mn: 40 g; Co: 0.1 g; Cu: 10 g; Se: 0.1 g; I: 0.5 g. WBT0: 0% replacement of cassava chips with winged bean tubers; WBT33: 33% replacement of cassava chips with winged bean tubers; WBT66: 66% replacement of cassava chips with winged bean tubers; WBT100: 100% replacement of cassava chips with winged bean tubers.

**Table 3 animals-13-00677-t003:** Effect of different roughage sources with cassava chips replaced by winged bean tubers on in vitro gas kinetics and cumulative gas production at 96 h after incubation.

Roughage Source	WBT Inclusion	Gas Kinetics ^1^			
		a	b	c	|a| + b
Rice straw, RS *(Oryza sativa)*	0	−18.33	137.3 ^b^	0.076	155.7 ^b^
	33	−14.94	130.9 ^b^	0.064	146.3 ^b^
	66	−13.19	125.4 ^b^	0.067	138.9 ^b^
	100	−12.33	130.6 ^b^	0.066	142.9 ^b^
Ruzi grass, RZ *(Brachiaria ruziziensis)*	0	−18.64	139.9 ^b^	0.077	157.5 ^b^
	33	−17.27	151.3 ^b^	0.077	168.9 ^b^
	66	−21.08	183.9 ^a^	0.074	204.9 ^a^
	100	−23.88	196.2 ^a^	0.079	220.1 ^a^
Napier grass, NP *(Cenchrus purpureus)*	0	−19.85	149.9 ^b^	0.076	168.7 ^b^
	33	−18.88	142.7 ^b^	0.087	162.2 ^b^
	66	−16.14	136.8 ^b^	0.081	152.9 ^b^
	100	−13.36	128.9 ^b^	0.076	142.3 ^b^
SEM		2.13	9.75	0.003	11.81
Roughage source	RS	−14.69 ^a^	131.1	0.068 ^b^	146.0
	RZ	−20.22 ^b^	145.6	0.077 ^a^	163.2
	NP	−17.06 ^a^	139.6	0.079 ^a^	156.5
Level of WBT	0	−18.94	142.4	0.077	160.6
	33	−17.03	141.7	0.076	159.2
	66	−16.80	131.1	0.074	145.9
	100	−16.53	129.8	0.074	142.6
Interaction	Roughage source × WBT level	0.07	*p* < 0.01	0.054	0.0113
Contrast	Rice straw	−14.69 ^b^	130.5 ^b^	0.068 ^b^	145.9 ^b^
	Grass	−18.63 ^a^	141.6 ^a^	0.078 ^a^	172.2 ^a^
Polynomial	WTB (Linear)	0.050	0.562	0.525	0.397
	WTB (Quadratic)	0.558	0.558	0.024	0.575
	WTB (Cubic)	0.937	0.823	0.027	0.863

^a,b^ Values in the same column with different superscripts differ according to *p* < 0.05 and *p* < 0.01. SEM = Standard error of the mean; WBT = Winged bean tubers. ^1^ a = The level of gas production from the immediately soluble fraction, b = the level of gas production from the insoluble fraction, c = the gas production rate constant for the insoluble fraction (b), a + b = the gas potential extent of gas production.

**Table 4 animals-13-00677-t004:** Effect of different roughage sources with casava chips replaced by winged bean tubers on in vitro degradability.

Roughage Source	WBT Inclusion	Degradability at 12 h (g/kg Dry Matter)			Degradability at 24 h (g/kg Dry Matter)		
		IVDMD	IVOMD	IVNDFD	IVDMD	IVOMD	IVNDFD
Rice straw, RS *(Oryza sativa)*	0	560.1	623.3 ^abcde^	623.3	604.5	630.2 ^de^	632.2
	33	582.8	593.3 ^bcde^	599.9	579.5	611.2 ^e^	625.1
	66	554.4	629.9 ^abcde^	606.8	572.4	630.8 ^de^	620.0
	100	546.5	546.4 ^e^	579.4	636.6	643.6 ^cde^	619.9
Ruzi grass, RZ *(Brachiaria ruziziensis)*	0	639.6	582.2 ^cde^	631.5	672.2	694.8 ^ab^	653.5
	33	618.5	625.1 ^abcde^	633.0	661.5	719.3 ^ab^	649.9
	66	616.1	675.6 ^ab^	617.9	664.9	681.7 ^bc^	663.3
	100	613.2	682.9 ^a^	636.8	638.8	693.2 ^ab^	635.9
Napier grass, NP *(Cenchrus purpureus)*	0	602.6	674.5 ^ab^	630.0	665.5	727.9 ^a^	651.6
	33	586.9	648.3 ^abc^	625.1	690.4	653.2 ^cd^	633.0
	66	598.7	552.2 ^de^	620.2	663.1	642.2 ^cde^	614.5
	100	604.2	639.9 ^abcd^	624.8	621.0	623.7 ^de^	610.8
SEM		21.1	25.7	15.1	20.7	12.3	16.6
Roughage source	RS	560.9 ^b^	598.2 ^b^	602.4	560.9 ^b^	628.9	624.3
	RZ	621.9 ^a^	641.5 ^a^	629.8	621.9 ^a^	697.2	650.6
	NP	598.1 ^a^	628.8 ^a^	625.0	598.1 ^a^	661.7	627.5
Level of WBT	0	600.8	626.7	628.3	647.4	684.3	645.7
	33	596.1	622.3	619.3	643.8	661.2	636.0
	66	589.7	619.2	614.9	633.5	651.6	632.6
	100	588.0	623.1	613.7	632.1	653.5	622.2
Interaction	Roughage source × WBT	0.87	<0.01	0.71	0.14	<0.01	0.17
Contrast	Rice straw	560.9 ^b^	598.2 ^b^	602.4 ^b^	598.2	628.9	624.3
	Forage	609.9 ^a^	635.1 ^a^	627.4 ^a^	659.7	679.5	639.1
Polynomial	WBT (Linear)	0.48	0.12	0.09	0.35	0.03	0.58
	WBT (Quadratic)	0.48	0.32	0.90	0.05	0.21	0.84
	WBT (Cubic)	0.46	0.13	0.36	0.57	0.41	0.97

^a–e^ Values in the same column with different superscripts differ according to *p* < 0.05 and *p* < 0.01. SEM = Standard error of the mean; WBT = Winged bean tuber. IVDMD: In vitro dry matter degradability; IVOMD: in vitro organic matter degradability; IVNDFD: in vitro neutral detergent fiber degradability.

**Table 5 animals-13-00677-t005:** Effect of different roughage sources with casava chips replaced by winged bean tubers on ammonia–nitrogen content (NH_3_-N) and total volatile fatty acids (TVFAs).

Roughage	WBT Inclusion	NH_3_-N			TVFA		
		4 h	8 h	Mean	4 h	8 h	Mean
Rice straw, RS *(Oryza sativa)*	0	16.6	19.1	17.9	65.4	80.0	72.7
	33	15.8	20.7	18.2	54.8	71.4	63.1
	66	15.8	21.2	18.5	72.8	66.8	69.8
	100	15.3	21.0	18.1	61.9	66.2	64.0
Ruzi grass, RZ *(Brachiaria ruziziensis)*	0	15.7	23.2	19.5	77.5	80.7	79.1
	33	15.2	19.7	17.5	77.3	85.1	81.2
	66	15.6	20.7	18.2	63.7	82.3	73.0
	100	17.2	21.3	19.2	66.9	80.3	73.6
Napier grass, NP *(Cenchrus purpureus)*	0	15.7	19.0	17.4	74.4	89.0	81.7
	33	15.2	20.6	17.9	71.8	81.8	76.8
	66	15.6	22.0	18.8	70.1	86.5	78.3
	100	15.7	21.7	18.7	67.8	72.0	69.9
SEM		2.34	1.88	1.25	4.99	3.77	3.03
Roughage source	RS	15.9	20.5	18.2	63.7	71.1 ^b^	67.4 ^b^
	RZ	15.9	21.2	18.6	71.3	82.1 ^a^	76.7 ^a^
	NP	15.6	20.8	18.2	71.0	82.3 ^a^	76.7 ^a^
Level of WBT	0	16.0	20.5	18.2	72.4	83.2 ^a^	77.8 ^a^
	33	15.4	20.3	17.9	68.0	79.4 ^a^	73.7 ^ab^
	66	15.7	21.3	18.5	68.8	78.5 ^ab^	73.7 ^ab^
	100	16.1	21.3	18.7	65.5	72.8 ^b^	69.2 ^b^
Interaction	Roughage source × WBT	1.00	0.96	0.91	0.18	0.16	0.31
Contrast	Rice straw	15.9	20.5	18.2	63.7 ^b^	71.1 ^b^	67.4 ^b^
	Grass	15.8	21.0	18.4	71.2 ^a^	82.2 ^a^	76.7 ^a^
Polynomial	WTB (Linear)	0.71	0.76	0.86	0.74	0.01	0.18
	WTB (Quadratic)	0.94	0.97	0.79	0.98	0.27	0.53
	WTB (Cubic)	0.89	0.56	0.92	0.02	1.00	0.06

^a,b^ Values in the same column with different superscripts differ according to *p* < 0.05 and *p* < 0.01. SEM = Standard error of the mean; WBT = Winged bean tuber. IVDMD: in vitro dry matter degradability; IVOMD: in vitro organic matter degradability; IVNDFD: in vitro neutral detergent fiber degradability.

**Table 6 animals-13-00677-t006:** Effect of different roughage sources with casava chips replaced by winged bean tubers on the proportion of volatile fatty acids in the rumen.

Roughage	WBT Inclusion	Acetate (mol/100 mol)			Propionate (mol/100 mol)			Acetate/Propionate Ratio			Butyrate (mol/100 mol)		
		4 h	8 h	Mean	4 h	8 h	Mean	4 h	8 h	Mean	4 h	8 h	Mean
**Rice straw, RS** * **(Oryza sativa)** *	0	66.5	64.2	65.4	24.1	27.6	25.8	2.8	2.3	2.5	9.5	8.2 ^ab^	8.8
	33	67.0	65.9	66.4	23.3	25.9	24.6	2.9	2.6	2.7	9.7	8.2 ^ab^	8.9
	66	67.8	66.5	67.1	22.5	24.7	23.6	3.0	2.7	2.8	9.7	8.9 ^a^	9.3
	100	67.4	68.0	67.7	22.3	23.2	22.7	3.0	2.9	3.0	10.3	8.9 ^a^	9.6
**Ruzi grass, RZ** * **(Brachiaria ruziziensis)** *	0	63.3	60.8	62.1	27.6	32.7	30.1	2.3	1.9	2.1	9.0	6.5 ^c^	7.8
	33	60.5	60.8	60.6	29.5	32.4	30.9	2.1	1.9	2.0	10.0	6.9 ^bc^	8.4
	66	64.0	63.0	63.5	27.3	29.0	28.1	2.3	2.2	2.3	8.7	8.0 ^ab^	8.4
	100	64.2	63.8	64.0	27.1	27.5	27.3	2.4	2.3	2.3	8.7	8.7 ^a^	8.7
**Napier grass, NP** * **(Cenchrus purpureus)** *	0	65.9	60.8	63.3	26.2	31.3	28.7	2.5	2.0	2.2	8.0	8.0 ^ab^	8.0
	33	65.0	61.5	63.2	27.2	30.6	28.9	2.4	2.0	2.2	7.8	7.9 ^abc^	7.8
	66	64.5	62.0	63.3	28.2	30.8	29.5	2.3	2.0	2.1	7.3	7.2 ^bc^	7.2
	100	63.3	64.9	64.1	29.1	28.0	28.6	2.2	2.3	2.3	7.7	7.0 ^bc^	7.3
**SEM**		1.17	1.09	0.68	1.31	1.27	0.74	0.19	0.14	0.10	0.43	0.41	0.31
**Roughage source**	RS	67.2 ^a^	66.1 ^a^	66.7 ^a^	23.1 ^b^	25.3 ^b^	24.2 ^b^	2.9 ^a^	2.6 ^a^	2.8 ^a^	9.8 ^a^	8.5 ^a^	9.2 ^a^
	RZ	63.0 ^b^	62.1 ^b^	62.6 ^b^	27.9 ^a^	30.4 ^a^	29.1 ^a^	2.3 ^b^	2.1 ^b^	2.2 ^b^	9.1 ^b^	7.5 ^b^	8.3 ^b^
	NP	64.7 ^b^	62.3 ^b^	63.5 ^b^	27.6 ^a^	30.2 ^a^	28.9 ^a^	2.4 ^b^	2.1 ^b^	2.2 ^b^	7.7 ^c^	7.5 ^b^	7.6 ^c^
**Level of WBT**	0	65.2	62.0 ^b^	63.6 ^b^	26.0	30.5 ^a^	28.2 ^a^	2.6	2.1 ^b^	2.3 ^b^	8.8	7.5	8.2
	33	64.2	62.7 ^b^	63.4 ^b^	26.7	29.6 ^a^	28.2 ^a^	2.4	2.2 ^b^	2.3 ^b^	9.2	7.6	8.4
	66	65.4	63.8 ^ab^	64.6 ^ab^	26.0	28.1 ^ab^	27.1 ^ab^	2.6	2.3 ^ab^	2.4 ^ab^	8.6	8.0	8.3
	100	65.0	65.6 ^a^	65.3 ^a^	26.1	26.2 ^b^	26.2 ^b^	2.5	2.5 ^a^	2.5 ^a^	8.9	8.2	8.5
**Interaction**	Roughage source × WBT	0.33	0.95	0.33	0.52	0.80	0.18	0.77	0.83	0.34	0.39	0.04	0.20
**Contrast**	Rice straw	67.2 ^a^	66.1 ^a^	66.7 ^a^	23.1 ^b^	25.3 ^b^	24.2 ^b^	2.9 ^a^	2.6 ^a^	2.8 ^a^	9.8 ^a^	8.5 ^a^	9.2 ^a^
	Grass	63.8 ^b^	62.2 ^b^	63.1 ^b^	27.8 ^a^	30.3 ^a^	29.2 ^a^	2.3 ^b^	2.1 ^b^	2.2 ^b^	8.4 ^b^	7.5 ^b^	8.0 ^b^
**Polynomial**	WTB (Linear)	0.50	0.03	0.03	0.31	0.03	0.01	0.52	<0.01	<0.01	0.23	0.14	0.08
	WTB (Quadratic)	0.70	0.95	0.72	0.82	0.97	0.77	0.90	0.86	0.80	0.64	0.90	0.75
	WTB (Cubic)	0.80	0.73	0.90	0.92	0.89	0.99	0.86	0.81	0.91	0.71	0.52	0.94

^a–c^ Values in the same column with different superscripts differ according to *p* < 0.05 and *p* < 0.01. SEM = Standard error of the mean; WBT = Winged bean tuber.

**Table 7 animals-13-00677-t007:** Effect of different roughage sources with casava chips replaced by winged bean tubers on bacterial, protozoal, and fungal zoospore populations in the rumen.

Roughage	WBT Inclusion	Bacteria, Log 10 Cell/mL			Protozoa, Log 10 Cell/mL			Fungal Zoospore, Log 10 Cell/mL		
		4 h	8 h	Mean	4 h	8 h	Mean	4 h	8 h	Mean
Rice straw, RS *(Oryza sativa)*	0	7.49	8.47	7.98	4.70	4.99	4.84	4.03	4.79	4.41
	33	7.72	8.08	7.90	4.45	4.82	4.64	4.21	5.00	4.60
	66	7.72	8.00	7.86	4.58	5.15	4.87	3.97	5.09	4.53
	100	7.74	7.99	7.86	4.81	5.04	4.92	4.09	4.57	4.33
Ruzi grass, RZ *(Brachiaria ruziziensis)*	0	7.84	8.40	8.12	4.40	5.06	4.73	3.90	4.81	4.36
	33	7.82	8.40	8.11	4.58	5.03	4.80	3.94	5.30	4.62
	66	7.86	8.07	7.97	4.49	4.99	4.74	3.85	5.12	4.48
	100	7.93	8.06	8.00	4.29	4.92	4.60	4.03	4.80	4.41
Napier grass, NP *(Cenchrus purpureus)*	0	7.86	8.21	8.04	4.41	5.16	4.79	4.07	4.82	4.44
	33	7.46	8.14	7.80	4.65	4.90	4.77	4.03	4.84	4.43
	66	7.87	8.21	8.04	4.62	5.14	4.88	4.02	5.17	4.60
	100	7.88	8.18	8.03	4.32	5.12	4.72	4.07	4.92	4.50
SEM		0.146	0.099	0.091	0.171	0.127	0.082	0.184	0.141	0.108
Roughage source	RS	7.67	8.13	7.90	4.64	5.00	4.82	4.08	4.86	4.47
	RZ	7.86	8.23	8.05	4.44	5.00	4.72	3.93	5.01	4.47
	NP	7.77	8.18	7.97	4.50	5.08	4.79	4.05	4.94	4.49
Level of WBT	0	7.73	8.36 ^a^	8.05	4.51	5.07	4.79	4.00	4.81 ^bc^	4.40
	33	7.67	8.20 ^ab^	7.93	4.56	4.91	4.74	4.06	5.04 ^ab^	4.55
	66	7.82	8.09 ^b^	7.95	4.56	5.09	4.83	3.95	5.13 ^a^	4.54
	100	7.85	8.08 ^b^	7.96	4.47	5.03	4.75	4.06	4.76 ^b^	4.41
Interaction	Roughage source × WBT level	0.53	0.39	0.47	0.47	0.76	0.23	0.99	0.40	0.72
Contrast	Rice straw	7.67	8.13	7.90	4.64	5.00	4.82	4.08	4.86	4.47
	Grass	7.81	8.21	8.01	4.47	5.04	4.75	3.99	4.97	4.48
Polynomial	WTB (Linear)	0.28	0.01	0.36	0.56	0.42	0.21	0.92	0.39	0.53
	WTB (Quadratic)	0.48	0.08	0.65	0.19	0.80	0.14	0.87	0.02	0.10
	WTB (Cubic)	0.71	0.58	0.99	0.72	0.13	0.12	0.36	0.44	0.79

^a–c^ Values in the same column with different superscripts differ according to *p* < 0.05 and *p* < 0.01. SEM = Standard error of the mean; WBT = Winged bean tuber.

## Data Availability

Not applicable.

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
