# Peer review of "In Vitro Evaluation of Winged Bean (Psophocarpus tetragonolobus) Tubers as an Alternative Feed for Ruminants"

_animals, 2023, doi:10.3390/ani13040677_

Round 1
Reviewer 1 Report
Does the experiment provide the positive control in the methodology part concerning for gas production?
Example of the trial of positive control is added with yeast or the things for gas production. This factor is could be better to be use as a case comparing on the performance between control group and sample test group about gas production.
Does the researcher certain that Psophocarpus tetragonolobus can replace to use with Cassava in the future? It would be perfect sentence if the authors mentions about the cost and yields comparing between "Winged bean" and "Cassava" in the introduction part.
Author Response
Response to Reviewer 1
Response: I would like to express my sincere gratitude for your helpful comment on this manuscript. Your suggestion to include information on cost and yield is invaluable, as it will make my article more complete and informative. I have taken your suggestion into consideration and have now included new data. I believe that this information will provide valuable insight into the potential for this crop as a feed source for ruminants. Once again, I would like to thank you for your constructive feedback and for taking the time to review my work. Your support and guidance are greatly appreciated.
Comments and Suggestions for Authors
Does the experiment provide a positive control in the methodology part concerning gas production?
Example of the trial of positive control is added with yeast or the things for gas production. This factor is could be better to be use as a case comparing on the performance between control group and sample test group about gas production.
Response: Thank you for your advice. We believe that having a positive control can better be used as a case comparing the performance of gas production and may reduce experimental variability by allowing comparison to a known standard. However, when studying a completely new and unknown feed, it may not be possible to have a positive control (especially using feed combinations between Roughage source and new feed source such as a WBT) as there is no prior knowledge or reference point. In these cases, the experiment focuses on finding novel feed or making observations that can provide the foundation for future alternative feed for ruminants. Furthermore, adding a positive control can also increase experimental variability as it requires additional handling, measurement, and manipulation, which can introduce new sources of error.
Does the researcher certain that Psophocarpus tetragonolobus can replace to use with Cassava in the future? It would be perfect sentence if the authors mentions about the cost and yields comparing between "Winged bean" and "Cassava" in the introduction part.
Response: Thank you for your suggestion. It is not possible to say with certainty whether Psophocarpus tetragonolobus can replace Cassava in the future. However, there are certain facts in the study's findings that support the potential of this. In addition, this includes the availability and accessibility of Psophocarpus tetragonolobus seed or tubers, the willingness of farmers to adopt this new crop, and the development of new technologies to increase its productivity and marketability. In conclusion, while it is not possible to say with certainty whether Psophocarpus tetragonolobus can replace Cassava, there are factors that suggest the possibility of this, and more research is needed to determine the feasibility of this alternative crop.
Regarding the reviewer's proposed price for the WBT, it would be great if there was more information to add to the introductory section. WBT prices, on the other hand, vary by region. There is no standard price to sustain it because it has not been broadly spread. Since it is a new commercial crop, normally this plant's pods are commonly consumed as food at a reasonable cost while its tuber has not been used as food before. As a result, citing any extra references to complete this text that the reviewer has previously recommended is problematic. While the production yield, the author has considered and added it to the introduction part as the reviewer has already recommended. please see the introduction (Line 71-76).

Reviewer 2 Report
Title: In vitro evaluation of Winged bean (Psophocarpus tetragonolobus) tubers as novel alternative feed for ruminants
Hypothesis: Winged bean tubers would be ideal for use as both animal feed and a novel feedstock for ruminants in the area due to the variable price of feed.
General notes:
1 – Standardize the formatting/spacing to denote significance throughout manuscript. Currently, formatting varies throughout (P<0.01, p<0.05, P < 0.05). Also, the “P” should be italicized throughout. The reviewer also noticed that only on 1 occasion did the authors reference a specific p-value, please revise throughout to include the specific values, not just P < 0.01.
2 – The authors use the following text throughout the manuscript: “winged bean tubers replacement cassava chip”. The meaning is incorrectly used as WBT are used to replace cassava chips. A correct way to phrase this is: “replacement of cassava chips with winged bean tubers ”. Correct throughout.
3 – Carefully review the manuscript for the inadvertent replacement of the word “chip” for “ship”.
4 – It is difficult to describe the required edits following Table 2 as lines are no longer labeled.
Specific notes:
Line 7 – Include address
Line 8 – Include address
Line 28/30 – Define abbreviation when first used
Line 42 – Should read “The problem with feed production”
Line 115 – Should read “In the morning, before feeding, rumen fluid”
Line 124 – Should read “being brought in from the laboratory within 15 minutes.”
Line 133/134 – Should read “blended in an anaerobic environment.”
Line 138 – Should read “contained butyl rubber stoppers with and aluminum seals crimped,”
Table 1 – Include units for each parameter, standardize number of decimal placings presented, and left justify text
Table 2 –Left justify text
Figure 1 – Remove decimal placing in units of y-axis, correct x-axis label to “Incubation time, h”
Section 3.3 – The following line missuses the words inconclusive and inefficient, please correct: “When assessing digestibility at 12 and 24 hours of incubation, the combination of WBT with NP grass was inconclusive, while the combination of RS and WBT was inefficient”
Figure 2 – Reposition p-diff and mean values away from the bars to improve visibility and correct x-axis label to “Incubation time, h”
Table 4 – Correct column headers to align with columns, left justify text, and consider changing “WBT ration” to “WBT inclusion or percentage”
Table 5 – Left justify text
Table 6 – Left justify text
Section 3.5 – Data in reference is from Table 7, not Table 6.
Section 4:
Remove “as shown in Figure 1”
Remove parenthesis from the sentence: “Olivera [33] showed that with a little amount of the easily soluble fraction in the feed), the utilization of the insoluble fraction was poor, resulting in a low nutritional value and a low Gas b.”
Should read “Akinfemi et al. [35] compared different”
Are repeating periods required for “n Olea europaea L.. and Citrus aurantium L. .”?
Should read “Negative Gas a values have also been”
Should read “no improvement in rice straw had reduced Gas c”
Revise line “Lignin, which is present in higher amounts in rice straw, Straw's high lignin content also contributes to its low digestibility ”
Revise “grass-basal diet” to “grass-based diet”
Revise “gas production are intimately related to” to “gas production is closely related to”
Revise “including the type of feed being intake.” to “including the type of feed being consumed.”
Revise “concentration in grass would be improve” to “concentration in grass would be improved”
Sentence should not begin with an acronym. Revise sentence: “RS not a highly digestible feed, and it is not a good source of energy for animals [48].” To “Rice straw is not a highly digestible feed or a good source of energy for animals [48].”
Define the meaning of “C3” when it is first used.
Author Response
Response to Reviewer 2
Title: In vitro evaluation of Winged bean (Psophocarpus tetragonolobus) tubers as novel alternative feed for ruminants
Hypothesis: Winged bean tubers would be ideal for use as both animal feed and a novel feedstock for ruminants in the area due to the variable price of feed.
Response: I am extremely grateful for the time and effort you have put into reviewing my article, "In vitro evaluation of Winged bean (Psophocarpus tetragonolobus) tubers as a novel alternative feed for ruminants." Your thorough and insightful suggestions for language adaptation have been incredibly helpful and I am truly impressed by your attention to detail. I have taken your suggestions into consideration and have made the necessary revisions to the manuscript. I am confident that the changes have greatly improved the clarity and impact of the article. Once again, thank you for your invaluable contributions to our work. Your expertise and guidance have been invaluable and I am honored to have had the opportunity to work with you on this project.
General notes:
1 – Standardize the formatting/spacing to denote the significance throughout manuscript. Currently, formatting varies throughout (P<0.01, p<0.05, P < 0.05). Also, the “P” should be italicized throughout.
Response: Thank you so much. As you mentioned, we change the P-value to uppercase and italics to achieve consistency throughout the manuscript, please see through the manuscript.
The reviewer also noticed that only on 1 occasion did the authors reference a specific p-value, please revise throughout to include the specific values, not just P < 0.01.
Response: Thank you for your suggestion; we have already revised specific values of P-value, please see through the manuscript.
2 – The authors use the following text throughout the manuscript: “winged bean tubers replacement cassava chip”. The meaning is incorrectly used as WBT are used to replace cassava chips. A correct way to phrase this is: “replacement of cassava chips with winged bean tubers ”. Correct throughout.
Response: Thank you so much, we agree with your comment and have revised it using the sentence “replacement of cassava chips with winged bean tubers” throughout this manuscript, please see through the manuscript.
3 – Carefully review the manuscript for the inadvertent replacement of the word “chip” for “ship”.
Response: Thank you very much, there are 8 cassava ship typos in this manuscript, we have checked and corrected every point as you suggested. Thank you for your critical review, please see through the manuscript.
4 – It is difficult to describe the required edits following Table 2 as lines are no longer labeled.
Response: Thank you for your suggestion; we have already edited and labeled descriptions that can be explained to readers in the footnote, please see on Table 2.
Specific notes:
Line 7 – Include address we have already
Response: Thank you so much, we have already included the address, please see in Line 7.
Line 8 – Include address we have already
Response: Thank you so much, we have already included the address, please see in Line 8.
Line 28/30 – Define abbreviation when first used
Response: Thank you very much, We realized RZ is a term whose entire name has never been described. As a result, we revised this section to make it more understandable to readers, please see Lines 28-30.
Line 42 – Should read “The problem with feed production”
Response: Thank you so much, we have already edited as you suggested, please see Lines 42.
Line 115 – Should read “In the morning, before feeding, rumen fluid”
Response: Thank you so much, we have already edited as you suggested, please see Lines 115.
Line 124 – Should read “being brought in from the laboratory within 15 minutes.”
Response: Thank you so much, we have already edited as you suggested, please see Lines 124.
Line 133/134 – Should read “blended in an anaerobic environment.”
Response: Thank you so much, we have already edited as you suggested, please see Lines 133/134.
Line 138 – Should read “contained butyl rubber stoppers with and aluminum seals crimped,”
Response: Thank you so much, we have already edited as you suggested, please see Lines 138.
Table 1 – Include units for each parameter, standardize number of decimal placings presented, and left justify text
Response: Thank you very much, we have already a standardized number of decimals to 1 unit and left justify text in the table, please see Table 1.
Table 2 –Left justify text
Response: Thank you so much, we have already edited as you suggested, please see Table 2.
Figure 1 – Remove decimal placing in units of y-axis, correct x-axis label to “Incubation time, h”
Response: Thank you very much, we have already edited as you suggested, please see Figure 1.
Section 3.3 – The following line missuses the words inconclusive and inefficient, please correct: “When assessing digestibility at 12 and 24 hours of incubation, the combination of WBT with NP grass was inconclusive, while the combination of RS and WBT was inefficient”
Response: Thank you very much, we believe that highlighting the relationship between RZ and WBT is sufficient to communicate with readers. We decided to remove the following sentence since it would cause confusion, please see line 276-278.
Figure 2 – Reposition p-diff and mean values away from the bars to improve visibility and correct x-axis label to “Incubation time, h”
Response: Thank you very much, we have already repositioned the word and changed x-axis labeled as you suggested, please see figure 2.
Table 4 – Correct column headers to align with columns, left justify text, and consider changing “WBT ration” to “WBT inclusion or percentage”
Response: Thank you so much, we have already left justify text and changed the word “WBT ration” to “WBT inclusion” in all table, please see through the manuscript.
Table 5 – Left justify text
Response: Thank you so much, we have already edited as you suggested, please see Table 5.
Table 6 – Left justify text
Response: Thank you so much, we have already edited as you suggested, please see Table 6.
Section 3.5 – Data in reference is from Table 7, not Table 6.
Response: Thank you so much, we have already edited as you suggested, please see section 3.5.
Section 4:
Remove “as shown in Figure 1”
Response: Thank you so much, we have already removed the sentence, please see Line 340.
Remove parenthesis from the sentence: “Olivera [33] showed that with a little amount of the easily soluble fraction in the feed), the utilization of the insoluble fraction was poor, resulting in a low nutritional value and a low Gas b.”
Response: Thank you so much, we have already removed parenthesis from the sentence, please see Line 371.
Should read “Akinfemi et al. [35] compared different”
Response: Thank you so much, we have already edited the sentence, please see Line 380.
Are repeating periods required for “n Olea europaea L.. and Citrus aurantium L. .”?
Response: Thank you so much, we have already edited the sentence, please see Line 385.
Should read “Negative Gas a values have also been”
Response: Thank you so much, we have already edited the sentence, please see Line 398.
Should read “no improvement in rice straw had reduced Gas c”
Response: Thank you so much, we have already edited the sentence, please see Line 415.
Revise line “Lignin, which is present in higher amounts in rice straw, Straw's high lignin content also contributes to its low digestibility ”
Response: Thank you so much, we have already revised the sentence, please see Line 431-433.
Revise “grass-basal diet” to “grass-based diet”
Response: Thank you so much, we have already edited the word, please see Line 441.
Revise “gas production are intimately related to” to “gas production is closely related to”
Response: Thank you so much, we have already edited the word, please see Line 443.
Revise “including the type of feed being intake.” to “including the type of feed being consumed.”
Response: Thank you so much, we have already edited the word, please see Line 450.
Revise “concentration in grass would be improve” to “concentration in grass would be improved”
Response: Thank you so much, we have already edited the word, please see Line 452.
Sentence should not begin with an acronym. Revise sentence: “RS not a highly digestible feed, and it is not a good source of energy for animals [48].” To “Rice straw is not a highly digestible feed or a good source of energy for animals [48].”
Response: Thank you so much, we have already edited the sentence, please see Lines 453-454.
Define the meaning of “C3” when it is first used.
Response: Thank you so much, we have already filled in the full name for the reader to understand, please see Line 464.
…………………………….Thank you!....................................................................................

Round 2
Reviewer 1 Report
If the authors still confirm that the Winged bean (Psophocarpus tetragonolobus) tubers could be used as a novel alternative feed for ruminants without comparing with the positive control. I would like to suggest that this test is could not be a "novel". However, it could be instead the word "an alternative feed". Moreover, the authors could find other previous evidence publications that provided information about positive control in an animal model (combination with feeds) related to gas production in the same field as the author's research topic without showing the error, in the case of the authors would like to improve the methodology design for future.
Author Response
Response to Reviewer 1
Comments and Suggestions for Authors
If the authors still confirm that the Winged bean (Psophocarpus tetragonolobus) tubers could be used as a novel alternative feed for ruminants without comparing with the positive control. I would like to suggest that this test is could not be a "novel". However, it could be instead the word "an alternative feed". Moreover, the authors could find other previous evidence publications that provided information about positive control in an animal model (combination with feeds) related to gas production in the same field as the author's research topic without showing the error, in the case of the authors would like to improve the methodology design for future.
Response: Thank you reviewer 1 for your suggestion. We appreciate your attention to detail as well as the time you took to provide such informative and valuable comments, and our article has been revised to reflect your recommendations. Now, the title​ has been​ changed to " In vitro evaluation of Winged bean (Psophocarpus tetragonolobus) tubers as an alternative feed for ruminants". Please​ see​ the​ manuscript.
Your recommendation for positive control points in future parts of the methodology will be particularly useful in ensuring the quality and accuracy of my work. We are also sincerely grateful for your encouragement and support in our writing journey!
…………………………………..Thank you!........................................................................
